# Understanding Drug Resistance of Wild-Type and L38HL Insertion Mutant of HIV-1 C Protease to Saquinavir

**DOI:** 10.3390/genes14020533

**Published:** 2023-02-20

**Authors:** Sankaran Venkatachalam, Nisha Murlidharan, Sowmya R. Krishnan, C. Ramakrishnan, Mpho Setshedi, Ramesh Pandian, Debmalya Barh, Sandeep Tiwari, Vasco Azevedo, Yasien Sayed, M. Michael Gromiha

**Affiliations:** 1Department of Biotechnology, Bhupat and Jyoti Mehta School of Biosciences, Indian Institute of Technology Madras, Chennai 600036, India; 2Protein Structure-Function Research Unit, School of Molecular and Cell Biology, University of the Witwatersrand, Johannesburg 2000, South Africa; 3Department of Genetics, Ecology, and Evolution, Institute of Biological Sciences, Federal University of Minas Gerais (UFMG), Belo Horizonte 31270-901, Brazil; 4Institute of Integrative Omics and Applied Biotechnology (IIOAB), Nonakuri, Purba Medinipur 721172, West Bengal, India; 5Institute of Biology, Federal University of Bahia, Salvador, BA 40110-909, Brazil; 6Institute of Health Sciences, Federal University of Bahia, Salvador, BA 40110-909, Brazil

**Keywords:** HIV protease subtype C, L38HL double insertion, Saquinavir, drug resistance, molecular dynamics simulations

## Abstract

Acquired immunodeficiency syndrome (AIDS) is one of the most challenging infectious diseases to treat on a global scale. Understanding the mechanisms underlying the development of drug resistance is necessary for novel therapeutics. HIV subtype C is known to harbor mutations at critical positions of HIV aspartic protease compared to HIV subtype B, which affects the binding affinity. Recently, a novel double-insertion mutation at codon 38 (L38HL) was characterized in HIV subtype C protease, whose effects on the interaction with protease inhibitors are hitherto unknown. In this study, the potential of L38HL double-insertion in HIV subtype C protease to induce a drug resistance phenotype towards the protease inhibitor, Saquinavir (SQV), was probed using various computational techniques, such as molecular dynamics simulations, binding free energy calculations, local conformational changes and principal component analysis. The results indicate that the L38HL mutation exhibits an increase in flexibility at the hinge and flap regions with a decrease in the binding affinity of SQV in comparison with wild-type HIV protease C. Further, we observed a wide opening at the binding site in the L38HL variant due to an alteration in flap dynamics, leading to a decrease in interactions with the binding site of the mutant protease. It is supported by an altered direction of motion of flap residues in the L38HL variant compared with the wild-type. These results provide deep insights into understanding the potential drug resistance phenotype in infected individuals.

## 1. Introduction

Human immunodeficiency virus (HIV) is the causative agent of acquired immunodeficiency syndrome (AIDS) and is responsible for millions of deaths worldwide. Particularly, the Asian and South African populations are among the most afflicted due to AIDS infections caused by non-B HIV subtypes, whose prevalence is lower in other parts of the globe [1,2,3,4]. As most of the anti-retroviral drugs in use today target proteins from the HIV subtype B, their efficacy against proteins from the HIV subtype C and other non-B HIV subtypes is quite limited [5,6,7]. Specifically, inhibitors of the HIV aspartic protease, including lopinavir, ritonavir, saquinavir, indinavir and amprenavir, exhibit a several-fold decrease in binding affinity to HIV subtype C in comparison with HIV subtype B [8,9]. This has led to an increase in drug resistance phenotypes emerging in HIV subtype C and a proportional increase in the recent prevalence of infections due to subtype C in America and Europe [10,11]. Therefore, it is of utmost need to understand the key structural differences between proteins of subtype B and subtype C and optimize existing drugs for repurposing and for the development of novel drugs with a broad spectrum of activity.

The HIV aspartic protease is one of the major drug targets of interest for anti-retroviral drug development. The protease of both subtypes B and C is composed of two monomers of 99 amino acids each, forming a C2-symmetric obligate homodimer [4]. The homodimer formation in HIV protease is necessary for the catalytic activity of the enzyme, as each monomer contributes to one-half of the substrate binding site. Despite the overall high structural similarity between the HIV protease of subtypes B and C, the HIV subtype C protease harbors eight distinct mutations in residue positions far from the substrate binding site in comparison with the HIV subtype B protease [12]. Specifically, the M36I mutation of HIV subtype C protease in the hinge region disrupts a critical salt bridge formed by residues E35 and R57, thereby increasing the flexibility of the hinge region [13]. The increased flexibility of the hinge region induces an increase in flexibility of the flap region, which indirectly affects proper substrate or inhibitor binding to the HIV protease active site, leading to drug resistance [14]. Similarly, several other primary and secondary drug resistance mutations have been identified in the HIV protease subtype C, which renders the existing drugs less effective in controlling the spread of AIDS infections at a global scale [15,16]. Recently, Ledwaba et al. [17] identified an HIV protease subtype C variant with the insertion of two residues at the 38th position (L38HL). The 3D structures of the wild-type and L38HL variants are shown in Figure 1A,B, respectively.

Molecular dynamics simulations facilitate the atomistic level understanding of molecular conformations over a time period, and it is widely used in the literature [18,19]. In this work, we performed molecular dynamics simulations for both wild-type and L38HL proteases in the presence of the protease inhibitor Saquinavir (SQV) to understand the drug resistance phenotype due to the insertion of L38HL at an atomistic level. The structure of SQV is shown in Figure 1C. Results from the molecular dynamics trajectory revealed that the L38HL variant exhibits an increase in flexibility at the hinge and flap regions, similar to the M36I mutation. Detailed analysis of intermolecular interactions with the inhibitor and binding free energy calculations showed that L38HL insertion decreases the affinity of the protease towards the inhibitor in comparison with that of the wild-type protease from HIV subtype C. Further, the insertion in the hinge region causes the flap tip to narrow, leading to a reduction in the interactions available for SQV. Moreover, PCA analysis revealed the altered direction of motion of the flap residues in the L38HL variant compared to the wild-type. Similarly, the insertion affects the correlation of residues in the dimer interface and flap dynamics.

## 2. Materials and Methods

### 2.1. Initial Coordinates of HIV-Protease Subtype C and the L38HL Variant

The initial coordinates of the South African wild-type HIV-protease subtype C were obtained from PDB (PDB ID: 3U71) [12]. The structures of wild-type HIV-protease subtype C and L38HL variant in complex with SQV were obtained by docking using AutoDock Vina [20], and this technique is commonly used in computational research [21]. Insights into the drug resistance mechanism due to the L38HL insertion were obtained by comparing the structural changes observed over the course of the MD simulation in two systems: (i) wild-type HIV-protease subtype C in complex with SQV, and (ii) L38HL variant in complex with SQV.

### 2.2. MD Simulation Protocol

The complexed forms of HIV-protease subtype C were simulated in an explicit solvent environment for 500 ns each using AMBERff14SB [22] force field available with the AMBER18 software package [23]. Prior to production MD runs, each of the systems was initially set by solvating the solute using TIP3P water models [24] and neutralizing the system using counter ions (Cl^−^). The water box was set such that no solute atoms were present within 10 Å from the boundary using the Particle Mesh Ewald (PME) [25] boundary condition. In addition to the above steps performed using the tleap module in the AMBER package, the protonation state of each amino acid residue was set appropriately. The initial energy minimization was performed with 1000 cycles of steepest descent and 1500 cycles of conjugate gradient (CG) methods by restraining the protein atoms, followed by 2500 cycles of CG-based unrestrained minimization of the whole system. In order to attain equilibrium, each system was heated from 0 K to 300 K with 500 ps long canonical MD using Berendsen temperature coupling [26]. The pressure of each system was then equilibrated to approximately 1 atm. Equilibration was extended for another 500 ps to ensure the overall equilibration process for 1 ns time period, which is essential to achieve the equilibrium (with potential energy fluctuation of ± 200 kcal/mol) to start the production run. All the solute atoms were restrained with a force constant of 100 kcal/mol-Å^2^ and 10 kcal/mol-Å^2^ during the temperature and pressure equilibrations, respectively. An integration time step of 2 fs was used and the snapshots were written for every 1 ps. All the bonds involving hydrogen atoms were treated using SHAKE constraints [27]. Since the protonation of Asp 25/Asp 25′ does not have a significant effect on the binding of the ligands, they were left unprotonated [13,28,29].

### 2.3. Trajectory Analysis

The global structural changes in each system were analyzed from the MD trajectory using the CPPTRAJ module from the AMBER package. Root-mean-square deviation (RMSD), root-mean-square fluctuation (RMSF) and radius of gyration (R_g_) were plotted as a function of simulation time to identify snapshots of the protein with significant conformational changes over the course of the simulation. Specifically, the conformational change observed in wild-type protease in complex with SQV was compared with the L38HL protease–SQV complex.

### 2.4. Dynamic Cross-Correlation

The conformational dynamics of the wild-type/L38HL protease bound to SQV were studied using a cross-correlation analysis. This analysis yields a cross-correlation matrix, $C_{i,j}$, which calculates the movement of one atom with respect to the other (Equation (1)).
(1)$$C_{i,j}=\frac{\langle ∆r_{i}\left(t\right).∆r_{j}{\left(t\right)\rangle }_{t}}{\sqrt{\langle \|∆r_{i}{\left(t\right)\|}^{2}\rangle {}_{t}}\sqrt{\langle \|∆r_{j}{\left(t\right)\|}^{2}\rangle_{t}}},$$
where the angular brackets indicate the time average over the simulation trajectory, $r_{i}$ and $r_{j}$ are the displacement vectors from the mean position of the α carbons in the i and jth residues, respectively. The matrix element $C_{i,j}$ ranges from −1.0 to 1.0. The value of −1.0 indicates a high negative correlation between residues i and j and vice versa [30].

### 2.5. Principal Component Analysis

Principal component analysis (PCA) helps to differentiate the essential motions of the protein residues from the local fluctuations. PCA is carried out in two steps: (i) extracting the essential motions from the translational and rotational motions, which is performed by fitting the trajectories to the reference frame and (ii) construction of a 3N × 3N covariance matrix (C) as given in Equation (2).
(2)$$C_{\left(i,j\right)}=\langle \left(x_{i}-\langle x_{i}\rangle \right)\left(x_{j}-\langle x_{j}\rangle \right)\rangle$$

The obtained matrix represents the deviation of atomic positions over the trajectory. The matrix is then diagonalized by an orthogonal transformation, as in Equation (3) [31].
(3)$$T^{T}\mathrm{CT}=\mathrm{diag}\;(\lambda_{1},\lambda_{2},\lambda_{3},\lambda_{4},\lambda_{5},\ldots .,\lambda_{N},);\lambda_{1},>\ldots >\lambda_{N}$$

The PCA analysis was carried out using the CPPTRAJ module in the AMBER package.

### 2.6. Binding Free Energy Calculations Using MM/GBSA

The Molecular Mechanics Generalised Born Surface Area (MM/GBSA) [32,33] method was used to calculate the binding free energies of SQV bound to the wild-type and L38HL forms of HIV-1 protease subtype C. The single trajectory approach was opted to compute the binding free energies of both systems using the MM/GBSA module in AMBER18. For each system, a total of 1000 snapshots with an interval of 200 ps obtained from the last 100 ns of the MD trajectories were used for energy calculations. In MM/GBSA, the binding free energy was calculated using Equation (4). Further, the enthalpy component is obtained from various contributions, including the van der Waals $\left(∆E_{vdw}\right)$, electrostatic $\left(∆E_{ele}\right)$, polar solvation ($∆G_{gb}$) and the non-polar solvation ($∆G_{SA}$). The polar solvation of each system is computed using the Generalised Born surface area method, and the non-polar solvation is calculated using the empirical method based on solvent accessible surface area.
(4)$$∆G_{bind}=∆G_{complex}-∆G_{receptor}-∆G_{ligand},$$
(5)$$∆G_{bind}=∆G_{pol}+∆G_{hydro},$$
(6)$$∆G_{pol}=∆G_{elec}+∆G_{gb},$$
(7)$$∆G_{hydro}=∆G_{vdw}+∆G_{SA}$$ and
(8)$$∆G_{SA}=\gamma ∆SASA+\beta .$$

For *β* and *γ*, the default values of 0.8 and 4.8 were used for empirical calculations. ∆SASA mentioned in Equation (8) was calculated using the linear combination of pairwise overlaps (LCPO) method [34].

## 3. Results and Discussion

From the simulation of the two systems, trajectories were obtained with coordinates written at 1 ps intervals for each system and aligned to their respective initial coordinates. The global structural changes in the wild-type/L38HL protease structures bound to SQV were analyzed using the root-mean-square deviation (RMSD), the radius of gyration (R_g_) and the root-mean-square fluctuation (RMSF).

### 3.1. Global Structural Stability

RMSD is primarily used to monitor fluctuations in the global structure of the protein between every successive pair of frames in a simulation trajectory. By comparing the RMSD variations between trajectories, the relative global structural stability can be visualized, and the effect of mutation on the global structure can be quantified. The comparison of RMSD values for wild-type and L38HL proteases in complex with SQV is shown in Figure 2A. Even though both the complexes were observed to maintain conformational stability during the initial 300 ns of simulation, the wild-type protease in complex with SQV exhibited a major fluctuation in structure between 300 ns and 400 ns of the simulation trajectory, with the RMSD varying between 1.5 Å and 3.8 Å. After 400 ns, the wild-type protease–SQV complex structure stabilizes with an RMSD of ~3.5 Å. On the other hand, RMSD of the L38HL mutant in a complex with SQV was found to maintain structural stability throughout the course of the simulation. In both wild-type and L38HL complexed with SQV, fluctuations were observed for a period of at least 100 ns in the simulation trajectory, indicating the structural changes in the protein induced by the presence of the inhibitor. Hence, the initial 100 ns data was not used for further analysis. Further, the structural changes quantified by RMSD variation seem to be more pronounced in the case of the wild-type in complex with SQV than the L38HL variant. This can be an indication of the reduced induction effect due to the binding of SQV with the L38HL variant, which is insufficient to induce pronounced conformational changes in the complex, thereby potentially leading to drug resistance.

The overall compactness of the wild-type and the L38HL variant was quantified using the radius of gyration (R_g_), given as the average distance of the Cα atoms of the protease from their center of mass over the equilibrated MD trajectories. The conformational preferences of both systems were characterized by the probability distribution of R_g_ (Figure 2B). The L38HL variant bound to SQV explores conformational states with the R_g_ varying between ~18 Å and ~19 Å resulting in a wider curve than the R_g_ of the wild-type, which varies between ~17.5 Å and 18.5 Å. The average R_g_ values for the wild-type and the L38HL variant are 17.75 ± 0.16 Å and 18.33 ± 0.20 Å, respectively, which revealed that the wild-type protease bound to SQV has a more compact structure than the L38HL variant.

### 3.2. Residue-Level Fluctuations in Protein Conformation

The root-mean-square fluctuation (RMSF) was calculated for all residues in each MD simulation trajectory to explain the conformational changes observed in each domain of the protein. Figure 3 shows the structure of the wild-type and L38HL variant colored based on the scaled RMSF values. On comparing the RMSF values of all residues from the wild-type and L38HL bound to SQV, the L38HL double insertion increases the conformational flexibility of the hinge residues in both monomers in comparison with the wild-type hinge regions.

Corresponding to the increase in hinge flexibility, a mild increase in flexibility of the flap regions of both monomers, which flank the binding site and enable inhibitor entry and exit, was also observed in the L38HL protease, which agrees well with the previous studies [14]. Further, increased flexibility was observed in the fulcrum region (only chain A) and the hinge region (both chain A and chain B) of the protease.

Conclusively, the RMSF calculations of both the wild-type and the L38HL variant show increased flexibility in the flap regions of the L38HL variant. Previous studies have indicated that an increase in flap flexibility due to mutations in the hinge region can prevent proper binding of inhibitors to the HIV protease active site, thereby potentially leading to drug resistance. Based on the RMSF values obtained from the MD simulation trajectory, it can be inferred that the insertion of residues in the hinge region may lead to drug resistance by preventing the proper binding of SQV, due to the enhanced hinge region flexibility.

### 3.3. Local Conformational Changes

The local structural changes caused due to the insertion were investigated in three directions: (i) the Val32–Val32′ Cα distances to quantify the size of the binding site, (ii) the Tri Cα angle to understand the flap dynamics and (iii) the Hinge Cα angle to understand the change in hinge dynamics upon insertion.

### 3.4. Distance between the Binding Site Residues

The change in the binding site volume has been quantified by measuring the distance between the flap tips and the flap to active site distances [35,36]. However, the insertion of His and Leu at the 38th position alters the coordinates of the flaps in the L38HL variant. Hence, the change in volume of the binding site was investigated by measuring the Cα distance between Val32 and Val32′. The probability distribution of the distance values is given in Figure 4C. For the wild-type distributions, the distance varied between 21–26 Å with a peak close to 23 Å, whereas for the L38HL variant, the peak lies around 25 Å and the distances are distributed between 22–27 Å. The mean and standard deviation of the wild-type distribution is 23.2 ± 0.85 Å, and that of the L38HL variant is 24.4 ± 0.845 Å. Hence, the changes in the distance distribution depict the increased binding site volume in the L38HL variant. This observation agrees with the radius of gyration analysis, where the wild-type protease is more compact than L38HL.

Further, the L38HL variant maintains a wider active site, making it more solvent exposed, thereby weakening the interactions of the inhibitor with the protein and significantly impacting the drug binding. Figure 4A,B show the representative snapshots for the average distance between Val32 and Val32′ in the wild-type and L38HL variant, respectively. The surface representation of the protein clearly shows the widening of the active site in the L38HL variant (Figure 4B).

### 3.5. Analysis of Flap Tri Cα Angles

The flap regions in HIV protease play a major role in the substrate/inhibitor entry and exit, where they establish interactions with the substrate/inhibitor. Hence, the interaction of the flap with the substrate/inhibitor plays a key role in drug susceptibility. Further, changes in the flap dynamics are shown to have a direct influence on drug resistance. The Tri Cα angle of the wild-type (chain A: residues 49–51 and chain B: residues 49′–51′) and L38HL variant (chain A: residues 51–53 and chain B: residues 51′–53′) was measured over the equilibrated trajectory to explore the change in the flap dynamics due to the insertion. The frequency distribution of the measured Cα angles is shown in Figure 5A,B.

The mean and standard deviation values of the Tri Cα angle are 106.95 ± 7.4°, 97.81 ± 9.7°, 93.87 ± 8.7°, 100.0 ± 9.8°, for wild-type chain A, wild-type chain B, L38HL chain A and L38HL chain B, respectively. From the analysis of Tri Cα angle, it can be observed that the flap angle of chain A is narrower in the L38HL variant than in the wild-type. However, for chain B, the angle follows a similar distribution to that of the wild-type. Wang et al. [36] reported that the narrowing of the flap tip gives rise to conformations that have altered protein–drug interactions, which lead to reduced drug susceptibility. The representative snapshot showing the average curling of the flaps of chain A and chain B in wild-type and L38HL variant bound to SQV is shown in Figure 5C,D.

### 3.6. Analysis of Hinge Tri Cα Angle

In order to understand the influence of the hinge dynamics on the flap, we analyzed the hinge angle for the wild-type and the L38HL variant. Unlike the flap tip angle, insertion at the 38th position did not alter the residues forming the hinge in the L38HL variant. The angle between residues 39–41 (chain A) and residues (39′–41′) for chain B were used to measure the hinge angle. The frequency distribution of the hinge angle is shown in Appendix A. The reduced frequency and the wider curve width of the L38HL variant suggest the increase in flexibility of the hinge compared with that of the wild-type. Further, a shift in the peak compared with the wild-type indicates the curling of hinges. Similar to the flap angles, the narrowing of the hinge angles is observed in the L38HL variant.

### 3.7. Principal Component Analysis of Wild-Type and L38HL Bound to SQV

The essential dynamics (a process of applying PCA to a protein trajectory) helps to understand the collective molecular motion of the entire protein, which are characterized by Eigen values and Eigen vectors of the covariance matrix $C_{\left(i,j\right)}$ (Equation (2)) and are represented with porcupine plots (Figure 6A,B). The direction of the needles in the plot corresponds to the predominant direction of motion of the Cα atoms of residues, and the length of the needles corresponds to the magnitude of the motion.

The porcupine plots obtained from the first principal component (PC) show the collective motion of the protease subunits of the wild-type (Figure 6A) and L38HL variant (Figure 6B). Since the contribution from the first PC obtained over the equilibrated trajectories is more than 50% (Appendix A) for both the wild-type and L38HL variant, the analyses corresponding to the first PC alone are shown.

Figure 6B shows that the L38HL insertion induces a change in the dynamics of the protein. This is observed from the altered motions of the Cα residues corresponding to L38HL compared with that of the wild-type. Precisely, the directionality of motion of the residues in the flap region of the L38HL tends to move away from the binding site. In contrast, for the wild-type, the residues of the flap regions tend to move towards the binding site. Apart from changes in the directionality of motion, the amplitude of the motion is also affected due to insertion, especially the insertion of the amino acid residues, which induces a mild change in the amplitude of motion in the residues lining the dimer interface. Thus, from the analysis of the essential dynamics, it can be concluded that, in comparison with the wild-type, the insertion of the residues in the hinge region alters the directionality of motion of the residues in the flap region and the residues at the homodimer interface.

### 3.8. Dynamic Cross-Correlation Analysis

Dynamic cross-correlation analysis is used to characterize the movement of atoms with respect to one another. Due to the computational complexity of the atomistic simulations, the cross-correlation, in general, is studied between the Cα atoms. The output of the cross-correlation analysis is an N × N heat map, where N is the number of Cα atoms in the protein. The dynamic cross-correlation matrix plotted as a heat map for the wild-type and L38HL protease is presented in Figure 6C,D, respectively. Regions R1 and R2 in Figure 6C,D comprise the hinge and the flap regions in chain A andchain B, respectively, while region R3 comprises the residues lining the dimer interface. Regions R1 and R2 exhibit a strong positive correlation in the wild-type, while the strength of the correlation is reduced in the L38HL variant. On the contrary, the residues around the active site exhibit a mild increase in positive correlation in the L38HL variant compared to that of the wild-type.

Interestingly, region R3 exhibits both positive and negative correlations in the wild-type, whereas the positive correlation is lost in the L38HL variant. In addition, we observed a positive correlation between the hinge of chain A and the cantilever of chain B as well as a decrease in a negative correlation between the R1 region and the C-terminal of chain A.

The results from PCA and the cross-correlation analysis clearly show that the insertion of two residues causes a perturbation in the internal dynamics of the protease affecting the directionality of motions and the correlated movements of the flap region and the residues located at the dimer interface owing to drug resistance.

### 3.9. Binding Free Energy Calculations

To assess the binding affinity of SQV with wild-type and the L38HL variant, MM/GBSA calculations were performed using the MM/GBSA module from the AMBER18 package. The binding free energy $\left(∆G_{bind}\right)$ of SQV with the wild-type and L38HL variant is −48.94 ± 4.63 kcal/mol and −45.81 ± 5.67 kcal/mol, respectively. The individual contributions from the polar and non-polar interactions are given in Table 1.

From Table 1, we observe that the hydrophobic interactions comprising van der Waals and non-polar solvation-free energies remain similar for both the wild-type and the L38HL variant and favor the binding of SQV with the protease complexes. However, significant changes are observed in the polar interactions. The polar interactions comprising the electrostatic and polar solvation-free energies possess a negative effect on the binding of SQV to the L38HL variant, thereby weakening the binding affinity. The enthalpy of L38HL is weakened by 3.13 kcal/mol relative to that of the wild-type. Nevertheless, it was overcompensated by the favorable polar solvation-free energy, which could be the result of increased solvent exposure of the ligand due to the opening of the binding site. To further understand the influence of the insertion on the protein-inhibitor interaction, a per-residue interaction network was obtained using the MM/GBSA module from AMBER18. Upon comparing the residue-level contributions in Figure 7A–D, it is evident that the lost interaction of SQV with the Asp25′ plays a crucial role in reduced drug stability. Additionally, the interaction graphs clearly show the loss of interaction of SQV with chain B of the L38HL variant. The same can be observed from the hydrogen bond interactions (Figure 8) as well.

The list of hydrogen bonds formed throughout the equilibrated trajectory is given in Table 2. We observed that SQV establishes interactions with both the chains of the wild-type protein. However, in the L38HL variant, the interactions with chain B are lost and make the inhibitor move toward chain A. Moreover, the number of hydrogen bonds formed by SQV with the wild-type is more than that of the L38HL variant. The considerable reduction in the number of hydrogen bonds in the L38HL variant allows a wide opening of the binding site.

Overall, comparing the intermolecular interactions between the wild-type and mutant protease in complex with SQV shows that there is a significant decrease in interactions with SQV observed in mutant protease due to the difference in flap conformation caused by the L38HL insertion. This can lead to the formation of a weaker protease–inhibitor complex, which might not be as potent as the wild-type protease-SQV complex, thus leading to drug resistance. The three-dimensional view of the SQV-protease interactions is also shown in Figure 8.

## 4. Discussion

HIV protease is one of the prominent targets for mitigating HIV infection [37,38]. To date, there are 10 different Food and Drug Administration (FDA)-approved HIV-1 protease inhibitors, namely Saquinavir (SQV), Indinavir (IDV), Ritonavir (RTV), Nelfinavir (NFV), Atazanavir (ATV), Tipranavir (TPV), Darunavir (DRV), Fosamprenavir (FAPV), Amprenavir (APV) and Lopinavir (LPV), which are used to mitigate viral infections [39]. However, the selection pressure, together with the dynamic nature of viral replication, gives rise to viral protease strains that can evade the effect of the protease inhibitors [40]. It has been reported that the HIV-1 subtype is most prevalent among the sub-Saharan African regions, and it accounts for 70% of the HIV infections caused globally [7,41,42,43]. Due to this reason, many mutations/insertions have been reported in the HIV-1 protease subtype C. Unlike mutations, insertions are rare events, even though they have become prevalent since 1999 [44,45]. They are positively correlated with the codons associated with resistance to protease inhibitors as well as decrease protease inhibitor susceptibility and improve viral fitness. Recently, a new type of HIV-protease subtype C called the L38HL variant is characterized by the insertion of His and Leu in the 38th position.

Computational analysis of the L38HL along with the wild-type HIV-1 PR type C bound to SQV revealed that the binding of SQV is more stable in the wild-type than the L38HL variant. The reduction in the affinity is mainly due to the wider active site conformation exhibited by the L38HL compared with that of the wild-type. Further, the analysis of flap tips showed a narrow tip region in L38HL than wild-type [36]. This narrowing of the flap tip demonstrates the existence of an early event in the flap-opening process in the apo HIV-1 protease [46]. The increased distance between the active site residues, together with the narrowed flap tips, leads to the loss of key interactions with the SQV [36]. This agrees with the per-residue interaction spectrum of SQV with the wild-type and the L38HL proteases, where it can be seen that the insertion leads to a loss of key interactions of SQV with the chain B. Precisely, the interactions with residues Asp25′, Ala28′ and Ile82′, which had a contribution of more than −1.0 kcal/mol in the wild-type, are lost in the L38HL variant. As indicated above, the wider active site conformation exhibited in the L38HL can be associated with the increased flexibility of the hinge and flap regions [35].

One reason for the increased flexibility in the flap regions could be due to the reduced hydrogen bond lifetime between the flap residues (Appendix A) in the L38HL variant. Further, the frequency distribution of intermolecular hydrogen bonds showed that the number of hydrogen bonds is more in the wild-type than in L38HL. Further investigation of the occupancy values of inter-chain hydrogen bonds revealed that, despite the loss of interactions in L38HL, in general, many hydrogen bonds showed increased occupancy values. Specifically, the occupancy value of the main chain oxygen atom from Leu5 that forms a hydrogen bond with the Arg186/190 side chain nitrogen (NH1) increased from 0.58 in wild-type to 0.81 in L38HL. However, several interactions with reduced occupancy were also observed in L38HL. Nevertheless, the overall increase in inter-chain hydrogen bond occupancy increases the rigidness of the C-terminal and the N-terminal regions. This agrees with our results from PCA, where the residues in the N-terminal and the C-terminal regions had a reduced magnitude of motion. As seen from the dynamic cross-correlation analysis, the decreased correlation could be the result of changes in the hydrogen bond occupancy.

Conclusively, it was observed from the simulations that the insertion of residues in the hinge region triggers conformational changes leading to increased binding site volume and narrowing of the flap tips [36]. These conformational changes can be the result of the sustained formation of certain hydrogen bonds together with the loss of certain hydrogen bonds in the dimer interface, which alters the conformation, resulting in reduced motion of residues in the N and C termini. Further, the change in hydrogen bond occupancy influences the correlated motion of residues inducing a change in the directionality of the motion of the residues. As a result of these conformational changes, SQV loses key interactions with the active site residues leading to drug resistance.

## 5. Conclusions

In this study, molecular dynamics simulations were used to understand and compare the conformations of wild-type and L38HL mutant HIV protease subtype C upon interaction with the protease inhibitor, Saquinavir (SQV). The structural stability and intermolecular interactions of both proteases were studied during the course of the simulation to evaluate the possibilities of the emergence of a drug-resistance phenotype due to the mutation. Interestingly, the L38HL variant was found to exhibit an increase in flexibility at the hinge and flap regions, similar to previous drug-resistant mutations reported in the literature [13,15,16]. Further, binding free energy values for wild-type–SQV and mutant–SQV complexes using the MM/GBSA method revealed that L38HL insertion decreased the affinity of the protease by 3 kcal/mol towards the inhibitor in comparison with that of the wild-type protease. Moreover, these insertions greatly influence the correlation of the residues and the directionality of motion of the flap residues leading to an increase in flap flexibility and a wide opening of the binding pocket. These results provide insights into understanding the potential drug resistance phenotype in infected individuals.

## Figures and Tables

**Figure 1 genes-14-00533-f001:**
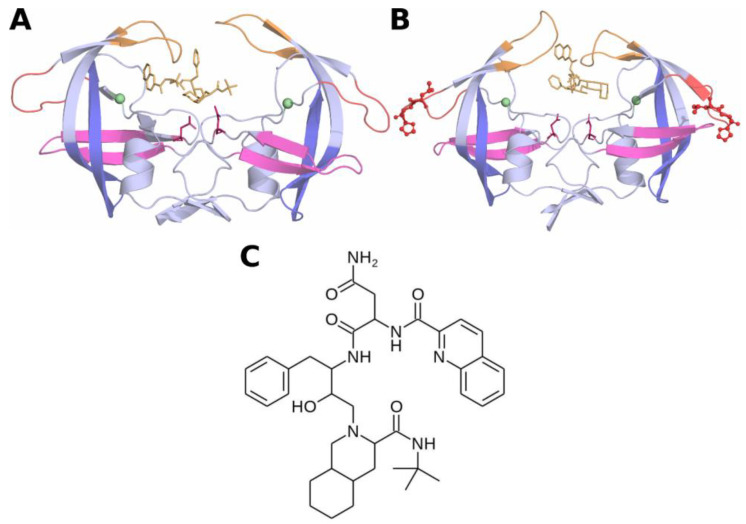
Structure of HIV-1 protease type C. (**A**) Wild-type and (**B**) L38HL. The protein is shown in cartoons, and the ligands are in the sticks. Insertion of residues His and Leu are represented in balls and sticks. The flap, hinge, cantilever and fulcrum regions are colored orange, red, blue and magenta, respectively. (**C**) 2D structure of Saquinavir (SQV).

**Figure 2 genes-14-00533-f002:**
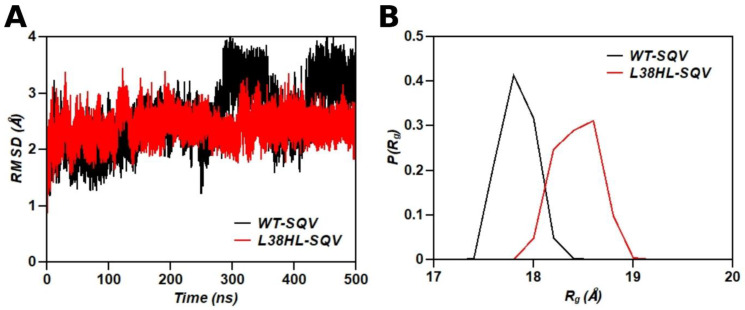
(**A**) RMSD plot from the MD compared the wild-type bound with SQV (black) and L38HL bound with SQV (red). (**B**) Probability distribution of radius of gyration (R_g_) was calculated for the equilibrated trajectory for the wild-type (black) and L38HL variant (red).

**Figure 3 genes-14-00533-f003:**
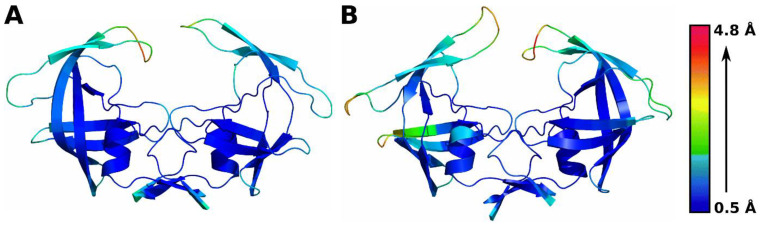
Root-mean-square fluctuations (RMSF) of Cα atoms of the protease. (**A**) wild-type bound to SQV. (**B**) L38HL variant bound to SQV. The coloring scheme was scaled based on the maximum and minimum values mentioned in the color bar. The blue color (minimum) indicates a low fluctuation, whereas the red color (maximum) indicates high fluctuations.

**Figure 4 genes-14-00533-f004:**
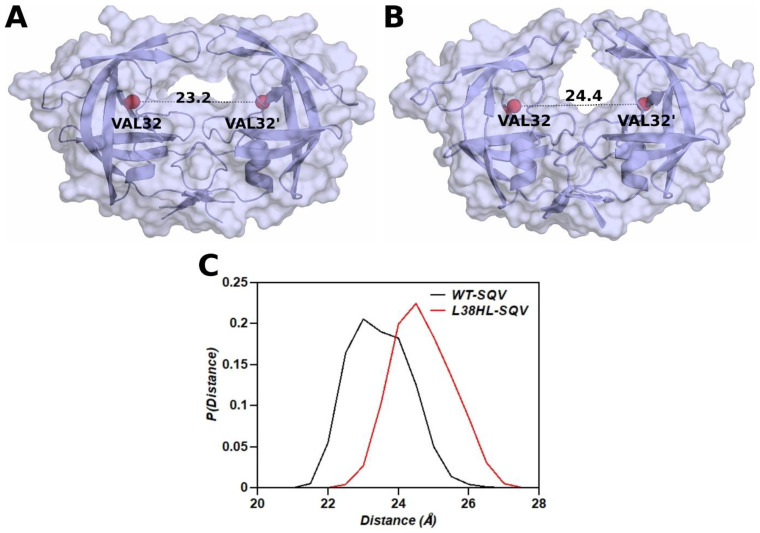
Representative snapshots showing the binding site distance measured between Val32 and Val32′ Cα (red sphere) in the (**A**) wild-type and (**B**) L38HL variant. Protein is represented in both the cartoon as well as surface representation. Ligand is not shown for visualization clarity. (**C**) Frequency distribution of Cα Val32–Val32′ in SQV bound WT and L38HL.

**Figure 5 genes-14-00533-f005:**
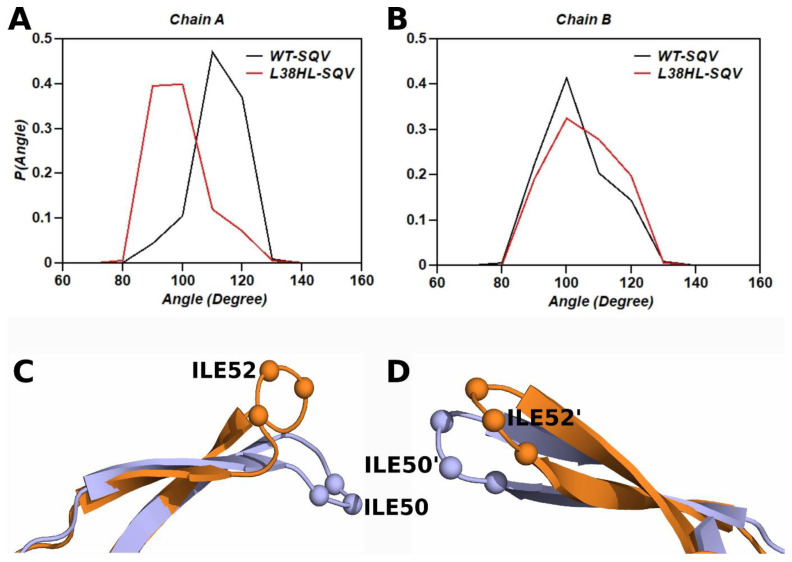
Frequency distribution of the Tri Cα angles in SQV-bound wild-type and L38HL complexes. (**A**) chain A (**B**) chain B. Representative snapshots of average flap Tri Cα angles corresponding to the wild-type (blue) and L38HL (orange) variant bound to SQV (**C**) Chain A (**D**) chain B. Chain A and chain B are independent snapshots.

**Figure 6 genes-14-00533-f006:**
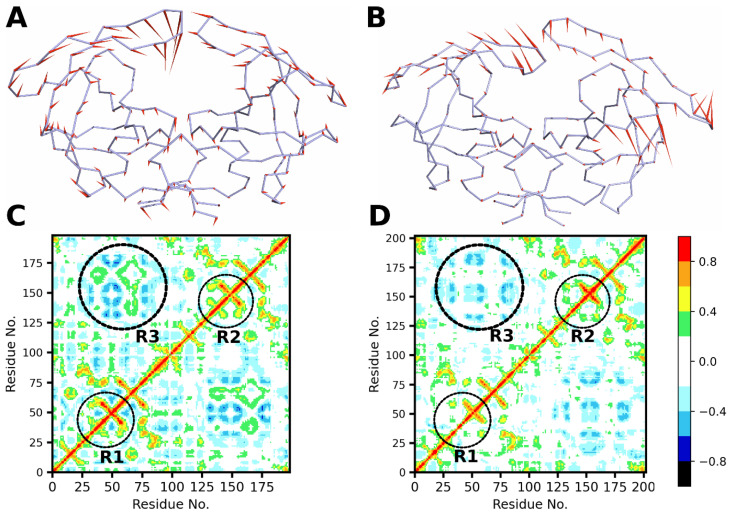
Stereo view of the porcupine plots showing the motion of the protease subunits from the first principal component from the (**A**) wild-type and (**B**) L38HL. Dynamic cross-correlation maps obtained of α-carbons for (**C**) wild-type and (**D**) L38HL. Residues that exhibit maximum correlation are colored red, and the residues that exhibit maximum anti-correlations are colored black. The values range from −1.0 to 1.0, where 1.0 indicates a positive correlation (red) among residues and −1.0 indicate a negative correlation (black). Regions R1, R2 and R3 (continuous numbering of residues is followed for clarity) represent hinge and flap regions in chain A, hinge and flap regions in chain B and residues at the dimer interface, respectively.

**Figure 7 genes-14-00533-f007:**
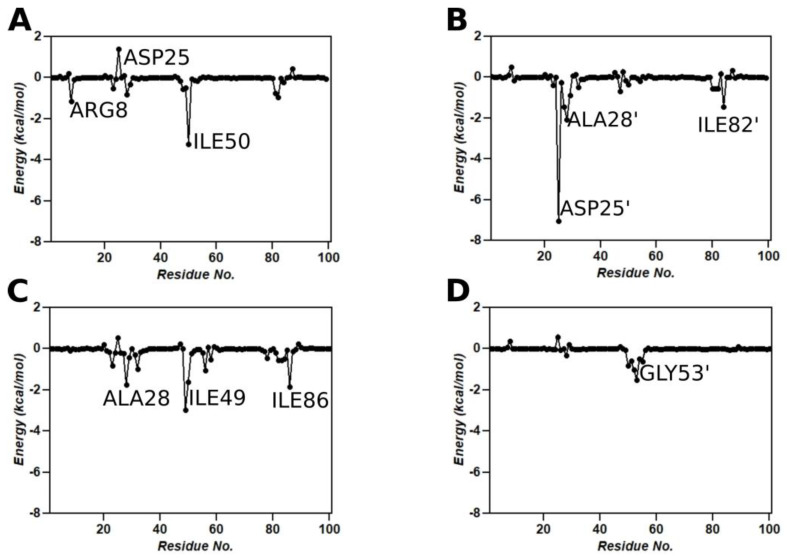
Per-residue interaction spectrum of (**A**) wild-type chain A, (**B**) wild-type chain B, (**C**) L38HL chain A and (**D**) L38HL chain B with SQV.

**Figure 8 genes-14-00533-f008:**
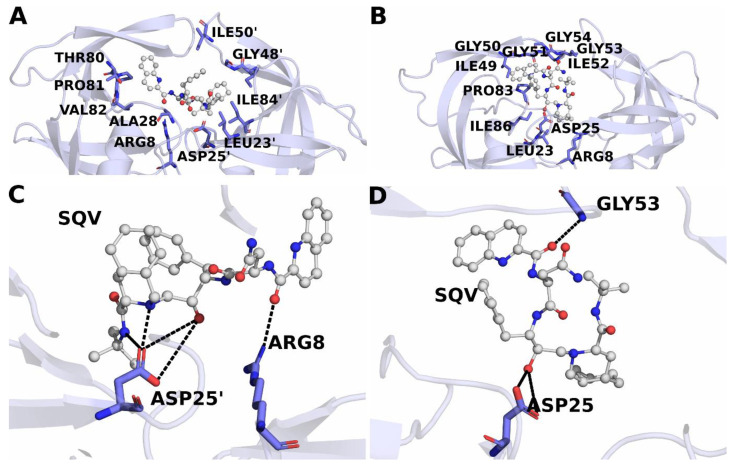
Snapshots that best represent the interactions between SQV with the (**A**) wild-type and (**B**) L38HL variant. The residues that lie within 4 Å of SQV are shown in (**A**) and (**B**). Further, the hydrogen bonding of SQV with the wild-type and L38HL variant is shown in (**C**) and (**D**), respectively. The hydrogen-bonded interactions are shown in black-dotted lines. SQV is shown in balls and sticks, and the residues forming interactions with SQV are represented as sticks.

**Table 1 genes-14-00533-t001:** Calculated binding free energy (kcal/mol) for wild-type and L38HL bound to SQV.

	Wild-type (kcal/mol)	L38HL (kcal/mol)
$∆G_{vdw}$	−57.22 ± 4.04	−58.10 ± 5.30
$∆G_{elec}$	−13.14 ± 12.66	35.11 ± 11.95
$∆G_{gb}$	29.31 ± 10.59	−15.10 ± 11.33
$∆G_{SA}$	−7.89 ± 0.45	−7.72 ± 0.56
$∆G_{pol}$	16.17 ± 11.62	20.01 ± 11.64
$∆G_{hydro}$	−65.11 ± 8.35	−65.82 ± 2.93
$∆G_{bind}$	−48.94 ± 4.637	−45.81 ± 5.67

All the reported values are in kcal/mol.$\;∆G_{bind}=∆G_{pol}+∆G_{hydro}$; $∆G_{pol}=∆G_{elec}+∆G_{gb}$ (Polar contribution); and $∆G_{hydro}=∆G_{vdw}+∆G_{SA}$ (Hydrophobic interactions).

**Table 2 genes-14-00533-t002:** List of hydrogen bonds formed by SQV with wild-type and L38HL sorted based on occupancy values.

	Acceptor	Donor	Average Distance (Å)	Angle (°)	Occupancy
wild-type	ASP124@OD1	SQV@N5	2.74	156.17	0.628
	SQV@O3	ARG8@NH1	2.81	160.26	0.51
	ASP124@OD1	SQV@N6	2.85	157.83	0.47
	SQV@O2	ASP128@N	2.86	162.61	0.25
	ASP124@OD2	SQV@N5	2.75	157.34	0.12
	ASP124@OD1	SQV@O4	2.75	162.47	0.10
	GLY126@O	SQV@N1	2.88	159.20	0.10
L38HL	ASP25@OD2	SQV@O4	2.63	164.89	0.45
	ASP25@OD1	SQV@O4	2.63	164.42	0.32
	SQV@O3	GLY53@N	2.86	157.20	0.19

## Data Availability

Data are available upon request.

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
