# Peer review of "Understanding Drug Resistance of Wild-Type and L38HL Insertion Mutant of HIV-1 C Protease to Saquinavir"

_genes, 2023, doi:10.3390/genes14020533_

Round 1

Reviewer 1 Report

The authors Sankaran et al, made an effort to understand drug resistance in HV1-C protease to saquinavir using molecular dynamics simulations. The authors using various measures show that the L38HL mutation exhibits an increase in flexibility at the hinge and flap regions. 

Overall the work is carried out well, and the paper is well-written.

I have one comment to make:

The authors state on page 4 "Principal component analysis (PCA), also known as the essential dynamics, helps to differentiate the essential motions of the protein residues from the local fluctuations"

I wish to point out the following from the reference the authors have given for PCA

1. Principal Component Analysis (PCA) is a multivariate statistical technique 

2. The process of applying PCA to a protein trajectory is called Essential Dynamics (ED).

Hence the authors cannot write as if PCA always implies essential dynamics.

The authors need to re-write this sentence to clear this misgiving.

Author Response

Reviewer 1

“The authors Sankaran et al, made an effort to understand the drug resistance in HIV1-C protease to saquinavir using molecular dynamics simulations. The authors using various measures show that the L38HL mutation exhibits an increase in the flexibility in the hinge and flap regions Overall, the work is carried out well, and the paper is well-written”.

Response: We thank the reviewer for the positive and constructive comments.

  1. The authors state on page 4 "Principal component analysis (PCA), also known as the essential dynamics, helps to differentiate the essential motions of the protein residues from the local fluctuations" I wish to point out the following from the reference the authors have given for PCA.

Principal Component Analysis (PCA) is a multivariate statistical technique 

The process of applying PCA to a protein trajectory is called Essential Dynamics (ED).

Hence the authors cannot write as if PCA always implies essential dynamics.

The authors need to re-write this sentence to clear this misgiving.

Response: We agree with the reviewer and the sentence has been modified appropriately in the revised manuscript.

Reviewer 2 Report

Authors have tried to understand the drug resistance of wild-type and L38HL insertion mutant of HIV-1 C protease to saquinavir. The study is designed well and conducted in a proper way. A few of my suggestions are mentioned below that can be considered. 

1. Abstract-methodology not clear.

2.  Introduction- the author can provide at least one sentence regarding molecular dynamics studies for a better understanding of readers. Author could check recently published studies: Homology model, molecular dynamics simulation and novel pyrazole analogs design of Candida albicans CYP450 lanosterol 14 α-demethylase. 

3. For docking study, author could check: Docking techniques in pharmacology: How much promising?. Computational biology and chemistry.

4. Discuss key findings with literature. 

5. Manuscript should be revised carefully for grammatical and typographical errors. 

Author Response

Reviewer 2

“Authors have tried to understand the drug resistance of wild -type and L38HL insertion mutant of HIV-1 C protease to saquinavir. The study is designed well and conducted in a proper way. A few of my suggestions are mentioned below that can be considered”.

Response: We thank the reviewer for the positive and constructive comments.

  1. Abstract-methodology not clear

Response: We have combined the information on different methods mentioned in the abstract and made it clear to the readers.

  1. Introduction- the author can provide at least one sentence regarding molecular dynamics studies for a better understanding of readers. Author could check recently published studies: Homology model, molecular dynamics simulation and novel pyrazole analogs design of Candida albicansCYP450 lanosterol 14 α-demethylase.

Response: we have included a sentence regarding molecular dynamics simulations in the introduction and cited the reference suggested by the reviewer.

  1. For docking study, author could check: Docking techniques in pharmacology: How much promising?

Response: We have cited the reference for docking.

  1. Discuss key findings with literature

Response: Key findings has been discussed along with other studies in the literature.

  1. Manuscript should be revised carefully for grammatical and typographical errors

Response: We have checked the manuscript for grammatical and typographical errors.